# The impact of ESG performance on corporate innovation—Empirical evidence from Chinese pharmaceutical listed companies

Liqiang Li[1], Su Wang[1,2]*, Yuwen Chen ᴰ [1,2]*

1 School of Business Administration, Shenyang Pharmaceutical University, Shenyang, China, 2 Drug Regulatory Research Base of NMPA, Research Institute of Drug Regulatory Science, Shenyang Pharmaceutical University, Shenyang, China

* wangsu414@163.com (SW); chenyuwen@syphu.edu.cn (YC)

## Abstract

Environmental, social and governance performance is an important driver for companies to achieve innovative and high-quality development, and the relationship between sustainable development and corporate innovation is one of the key concerns of pharmaceutical companies. This study aims to reveal the intrinsic role mechanism of sustainable transformation of pharmaceutical enterprises in high-quality innovation development. Based on the relevant data of China's A-share pharmaceutical listed companies from 2015 to 2022, the study establishes a fixed effect model and a mediation effect model, empirically examines the impact of ESG performance on the innovation of pharmaceutical listed companies, and deeply explores the role mechanism of R&D personnel investment and government subsidies in it. The study finds that: (1) ESG performance significantly enhances firms' innovation inputs and green innovation outputs; (2) this facilitating effect is more substantial among non-state-owned firms and firms in the eastern region; (3) good ESG performance has a more pronounced facilitating effect on innovation inputs of large firms, while it has a more prominent effect on the green innovation outputs of small and medium-sized firms; and (4) the mechanism analysis indicates that internal R&D personnel investment and external government subsidies play a significant mediating effect between ESG performance and corporate innovation. This study provides new empirical evidence for understanding the mechanism of ESG performance on corporate innovation in Chinese pharmaceutical companies. It provides theoretical support for proactively promoting the sustainable transformation and deepening the development of green innovation in Chinese pharmaceutical companies. In addition, there are important policy implications for promoting the sustainable transformation of pharmaceutical companies: the establishment of differentiated ESG incentive policies, the improvement of R&D talent training mechanisms, and the optimization of government

**Data availability statement:** All relevant data are within the manuscript and its Supporting information files.

**Funding:** The author(s) received no specific funding for this work.

**Competing interests:** The authors have declared that no competing interests exist.

subsidy allocations, focusing on supporting green innovation by companies with excellent ESG performance.

## Introduction

Under the general trend of coordinated development of the global economy and environment, enterprises worldwide have embarked on the road of sustainable development. As one of the key indicators to evaluate the sustainable development level of enterprises, the performance of corporate environmental, social and corporate governance (ESG) has not only received extensive attention in the international field, but also attracted much attention in China. The development of China's manufacturing industry generally shows the problems of high energy consumption, high pollution and low resource utilization [1], which puts tremendous pressure on the ecological environment. In terms of policy, the Chinese government attaches great importance to the sustainable development of enterprises and has given much support. For example, in May 2021, China's Ministry of Ecology and Environment (MEE) issued the "Reform Plan for the Legal Disclosure of Environmental Information", which plans to establish a basic mandatory disclosure system for environmental information by 2025 [2]. In June of the same year, the China Securities Regulatory Commission (CSRC) revised the format and content guidelines for annual and semi-annual reports of listed companies to add a section on "environmental and social responsibility" [3,4].

As a key industry related to human health and social well-being, the pharmaceutical industry's ESG performance not only affects corporate reputation and long-term competitiveness, but may also profoundly influence the direction and efficiency of corporate innovation. In recent years, governments, investors and consumers have put higher demands on the ESG practices of pharmaceutical companies. In 2021, China's Ministry of Industry and Information Technology and nine other departments jointly issued the '14th Five-Year Plan' for the development of the pharmaceutical industry [5], with the overall requirements of accelerating the innovation-driven development and transformation, and promoting the high-end, intelligent and green industry. ESG practices in Chinese pharmaceutical companies are more complex due to their industry specificity [6]. This study considers the pharmaceutical industry an ideal setting to observe how ESG and innovation work together. First, the environmental dimension in ESG drives firms to optimize the ethical path of technology from the early R&D stage by constraining resource consumption and pollution emissions [7]. Second, substantial compliance with global drug regulation internalizes social responsibility as a barrier to entry for innovation [8], and companies need to respond to ESG's social demands by improving access to medicines [9,10], such as affordable pricing and rare disease drug development, to reduce regulatory risk and gain policy support. Finally, the risk of governance imbalance faced by high market concentration and patent barriers highlights that governance mechanisms under the ESG framework, such as open innovation platforms and patent pool sharing, can reshape the industry's innovation ecosystem, prompting firms to balance their

commercial interests with public health objectives [11,12]. The logic of the pharmaceutical industry's long R&D cycle and strict regulation amplifies the normative role of ESG on technology ethics and resource allocation and provides an empirical basis for the theoretical construction of the interaction mechanism between ESG and innovation. Combined with the Chinese government's emphasis on the sustainable transformation of China's manufacturing enterprises, pharmaceutical companies, as high-tech enterprises in the manufacturing industry, are among the most important subjects for sustainable transformation. Therefore, this study explores whether the active disclosure of ESG performance by pharmaceutical listed companies will positively impact the sustainable transformation process.

Existing ESG-related studies have mainly focused on the impact of ESG factors on the short-term economy, such as enhancing corporate financial performance [13–15] and enterprise value [16–21], and the impact on corporate sustainable development is relatively rare, especially the research on its impact on corporate innovation is scarce. Green innovation is an important driving force for low-carbon transformation and sustainable development of enterprises [22], and the performance of green innovation is affected by ESG performance [23]. Existing studies have focused on the impact of the external environment, corporate structure and other factors on green innovation. For example, green credit policies can promote green innovation performance and quality of firms [24], and environmental regulations and foreign direct investment (FDI) inflows have a positive impact on green technological innovation [25]. Regarding corporate governance, the more positive the management tone, the more it promotes green innovation in the firm [26]. ESG performance is an input for companies to pursue sustainable development, and green innovation is a key driver of sustainable development for all companies and listed pharmaceutical companies. Therefore, an in-depth discussion of the impact of ESG performance on innovation in Chinese pharmaceutical companies is essential. What is the relationship between ESG performance and green innovation among listed Chinese pharmaceutical companies? Does this relationship vary according to firm characteristics? What are the underlying mechanisms? Sorting out these issues will help stimulate the enthusiasm for innovation in China's listed pharmaceutical companies and promote the rapid transformation of their green innovation achievements. However, it will also drive them to achieve high-quality development.

In summary, to address the above issues, this paper adopts the theoretical analysis method and empirical research method to take A-share pharmaceutical listed companies in 2015~2022 as the research samples, establishes the fixed effect model and the mediation effect model, and analyses and discusses the empirical results to validate the relationship between ESG performance and green innovation of pharmaceutical listed companies. In addition, the role of R&D personnel investment and government subsidies in their relationship is further analyzed and discussed. The study data and indicators were also grouped and regressed according to the nature, size and regional differences of Chinese pharmaceutical listed companies, to compare and analyze the impact of ESG performance on corporate innovation between large pharmaceutical listed companies and small and medium-sized pharmaceutical listed companies, as well as the differences in the impact of ESG performance on corporate innovation between pharmaceutical listed companies in the eastern, central and western regions.

The significance of this paper lies in the following: ① At the theoretical level, this paper enriches the literature on ESG manifestations in the field of green innovation, elaborates the value of ESG in guiding the sustainable development of Chinese pharmaceutical enterprises from a non-economic perspective, and provides a reference for subsequent related studies; ② At the practical level, based on the background that the government encourages pharmaceutical enterprises to innovate, this paper clarifies the influence mechanism of ESG performance on the green innovation of pharmaceutical listed enterprises, enhances the importance of ESG performance of enterprises, deepens the knowledge of enterprises on the value of ESG performance, and sheds light on the green innovation management of enterprises, and provides theoretical support for the sustainable development of enterprises.

## Theoretical basis and research hypothesis

### ESG performance and enterprise innovation

ESG is the Environment, Social and Governance acronym, an investment concept and enterprise evaluation standard that focuses on enterprises' environmental, social and governance performance instead of financial performance. ESG

evaluation is the quantitative scoring or rating of an enterprise's specific performance in the three dimensions of environment, social responsibility and corporate governance by international leading or Chinese related ESG evaluation organizations on the enterprise's ESG implementation from the three dimensions of environment, social responsibility and corporate governance, through the gradual refinement of the three dimensions of the indicators.

Enterprise innovation is based on the development trend of market demand. To produce and operate products that meet market demand, enterprises must make full use of and constantly optimize their own resources and social resource allocation, from the enterprise management at all levels of creation and innovation [27]. Pharmaceutical enterprise innovation is an important way to cope with the challenges of aging and improve public health. On the one hand, emergencies such as the COVID-19 epidemic have highlighted the importance of pharmaceutical innovation, and improving corporate innovation capacity can help to respond quickly to future challenges. On the other hand, an aging society has increased demand for chronic and geriatric treatments, and innovative medicines and therapies can better meet these needs.

ESG performance in terms of innovation investment. Innovation investment has a specific 'medicine' effect on ESG performance and high-quality enterprise development [28]. Investment in innovation is a fundamental catalyst for enhancing enterprises' productivity and fostering long-term economic growth [29]. On the one hand, depending on resource allocation theory, ESG performance gives more attention to the performance of enterprises in terms of environmental friendliness, fulfilment of social responsibility and sustainable development, so bridging the gap in the disclosure of the performance of enterprises in relation to society, shareholders, suppliers, employees and other stakeholders. The ESG investment framework aligns with energy conservation and emissions reduction objectives. Investors can assess their investment behavior and social responsibility contributions by analyzing company social responsibility performance; strong ESG performance will enhance investor trust and investment propensity. This will incentivize corporations to regulate carbon emissions and advance sustainable development efficiently. Moreover, good ESG performance means that pharmaceutical companies use raw materials that meet environmental standards, save energy in all aspects of the production process, and, in turn, produce green products [30]. Consequently, from inception to the conclusion of production, organizations will be compelled to augment their expenditure in research and development and enhance resource utilization efficiency. On the other hand, according to signaling theory, the disclosure of high-quality social responsibility information can communicate to society that enterprises prioritize environmental protection, social responsibility, and corporate governance, thereby bolstering investor confidence in the sustainable and robust development of enterprises, which in turn encourages them to pursue green scientific and technological innovation [31]. Companies that prioritize socially responsible performance are more likely to have broad and strong relationships with multiple stakeholders, and these networks provide access to diverse innovation resources and an expedited innovation process [32].

ESG performance in terms of innovation outputs. According to the tenets of stakeholder theory, investors and consumers have demonstrably elevated their predilection for green products and services [33], and green innovation outputs are inevitably required to satisfy the social interests of business stakeholders. China's economic development today has gradually shifted from a stage of high growth at the expense of the environment to a stage of high-quality development. Therefore, green innovation is one of the most proactive methods for businesses to reap the rewards of environmental growth. Thus, putting ESG principles into practice, encouraging green technology innovation, and developing green products are essential for businesses to achieve sustainable development goals [34]. Numerous researchers have established that a company's ESG rating positively influences its green technological innovation and that an increase in the number of green invention patents may also be attributed to ESG performance [35–37]. Furthermore, enterprises in sensitive sectors within developing nations exhibit enhanced environmental performance, and ESG practices enhance the study and implementation of corporate sustainable management [38–41].

Based on the analysis of resource allocation theory and signaling theory, ESG performance is favorable to the innovation input of pharmaceutical companies. Based on stakeholder theory and existing research discussions, ESG

performance favors the green innovation output of pharmaceutical companies. Therefore, this study proposes the following research hypotheses:

Hypothesis 1a (H1a). Firms' ESG performance is advantageously correlated with innovation investment, with better ESG performance helping firms increase innovation input.

Hypothesis 1b (H1b). Firms' ESG performance is advantageously correlated with innovation output, with better firms' ESG performance contributing to higher innovation output.

## ESG, R&D personnel and enterprise innovation

Alongside research funding, R&D talent is a strategic asset that propels innovation-driven growth and strengthens regional innovation capacities. Corporate social responsibility can communicate a company's beliefs and standards to the talent market, recruiting a greater pool of high-caliber research and development professionals. Augmenting proficient R&D personnel might elevate research efficiency, enhancing the enterprise's total innovation and research capabilities [42,43]. In addition, improved CSR performance increases the stability of a firm's R&D team; the more inventor managers and R&D personnel in the R&D team, the more positively team stability affects firm innovation [44].

The impact of R&D personnel on firms' innovation begins with improving the training of R&D personnel. This enhances staff members' capacity for technological innovation, helps them learn new concepts in scientific research and technology development, and helps them become proficient in the tools and techniques of technology development—all of which support the advancement of enterprise innovation [45]. Second, the company's R&D staff can develop team morale, increase productivity, and successfully convert into innovative output by engaging in various social responsibility events. They can also obtain useful experience and practical benefits [46]. Employee development practices help improve firms' performance, and as a result, firms may have more money to invest in business innovation [47]. In China's Yangtze River Delta region, R&D personnel mobility affects firms' innovation performance through direct and spatial spillovers [48]. A study identified an inverted U-shaped correlation between the stability of R&D teams and the quality of innovation [49], and R&D team stability significantly and positively affected firm innovation performance [50].

In conclusion, within the framework of robust internal controls, R&D people are positively affected by indicators of strong ESG performance and supply essential foundational resources for green innovation initiatives. Consequently, this study posits the subsequent research hypotheses:

Hypothesis 2a (H2a). Innovation investment is positively correlated with R&D personnel, who mediate between firms' ESG performance and innovation input.

Hypothesis 2b (H2b). Innovation output is positively correlated with R&D personnel, who mediate between firms' ESG performance and innovation output.

## ESG, government subsidies and enterprise innovation

Companies' ESG performance facilitates their access to government subsidies. In contrast to industrialized nations, political capital is a significant resource in the operations of Chinese enterprises [51]. In Asia, numerous government-backed investment funds participate heavily in ESG operations, demonstrating the significance of ESG principles for societal development [52]. Companies' commitment to environmental stewardship and transparency in environmental reporting presents a favorable image of environmental professionals to the global community. Companies can obtain additional government subsidies and attract investors that offer financial support for research and development.

Companies' access to government subsidies is conducive to further innovation. From a signaling perspective, government subsidies can convey a favorable indication that the government acknowledges a firm's capacity for innovation and possesses confidence in the industry's future, thereby mitigating information asymmetry through innovation [53]. It also helps firms with severe financing constraints to obtain credit financing and enhances their confidence in scientific and

technical innovation. From the resource viewpoint, the linkage between the government and enterprises will affect the green technology innovation activities of enterprises, the government subsidy policy has an important role in promoting the innovation activities of enterprises, and the promotion of China's green technology innovation cannot be separated from the power of the government [54]. Other studies reveal that government subsidies exhibit an inverted U-shaped effect on firm viability, initially promoting viability and then decreasing as government subsidies intensify [55]. Green innovation in pharmaceutical companies is long-term and risky [56], severely limiting companies' desire and trust in making green innovation decisions. In contrast, companies that maintain a good relationship with the government can obtain more government support and share innovation risks and losses [57], thus having the confidence to engage in green innovation activities.

In summary, ESG performance can enhance government-business interactions and allocate additional resources for green innovation, fostering innovation. This study presents the subsequent research hypotheses:

Hypothesis 3a (H3a). Innovation input positively correlates with government subsidies; government subsidies mediate between firms' ESG performance and innovation input.

Hypothesis 3b (H3b). Innovation output correlates positively with government subsidies; government subsidies mediate between firms' ESG performance and innovation output.

The analytical framework of the research is shown in Fig 1.

## Research design

### Sample selection and data source

This paper selects Chinese A-share listed pharmaceutical manufacturing companies based on the 2012 edition of the China Securities Regulatory Commission (CSRC) industry classification guidelines as the research object. CSI has been assessing the ESG performance of China's A-share and debt issuers since 2009, and now covers all A-share listed companies. The industry and academics have widely recognized the index [58]. Therefore, this paper adopts the CSI ESG database set for Chinese firms to reflect the ESG performance of the sample firms. Due to the emerging concept of ESG investment and the short period for wide-scale disclosure by rating agencies, this paper combines data availability and lag period needs to set the sample period from 2015 to 2022. After the following data processing steps: first, we eliminated samples of the ST (Companies with special treatment due to two consecutive years of operating losses) and *ST (Companies with delisting warnings due to three consecutive years of operating losses) types, then eliminated samples with missing data, and finally apply 1% and 99% tailing to continuous variables.

We acquire 1635 sample observations in an unbalanced panel data set. The ESG information is sourced from the Wind Information Financial Database (https://www.wind.com.cn/), the China Research Data Service Platform (CNRDS, https://

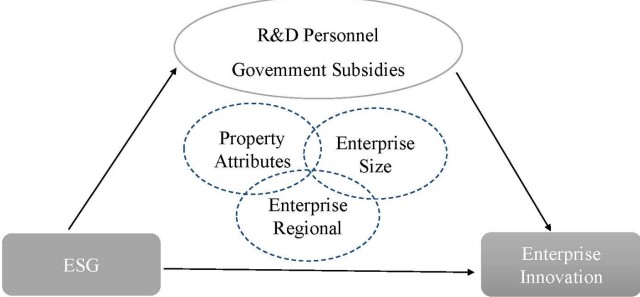

**Fig 1. Analytical framework.** The content within the grey rectangular box constitutes the core variables of the study. The elliptical content represents the mediating variables of the research. The dashed ellipse denotes the research perspective employed for heterogeneity analysis.

www.cnrds.com/) provides the green innovation patent data used for corporate innovation, and the China Stock Market & Accounting Research Database (CSMAR, http://cndata1.csmar.com/) provides all other information.

## Quantitative model

**Baseline regression model.** To evaluate the above hypothesis, the subsequent high-dimensional fixed effects model is constructed to analyze the influence of ESG performance on company innovation:

$$RD_{it} = \alpha_0 + \alpha_1 ESG_{it} + \alpha_2 Controls_{it} + City_i + Year_t + \varepsilon_{it} \tag{1}$$

$$Patent_{it} = \beta_0 + \beta_1 ESG_{it} + \beta_2 Controls_{it} + City_i + Year_t + \varepsilon_{it} \tag{2}$$

Where i is the firm; t is the year; $RD_{it}$ is the firm's innovation inputs, $Patent_{it}$ is the firm's green innovation outputs, and $ESG_{it}$ is the firm's ESG rating performance. To mitigate misleading results from omitted variables, a set of control variables related to financial performance is added, denoted by $Controls_{it}$. $City_i$ represents fixed effects in a territory, whereas $Year_t$ represents fixed effects over a year. The random disturbance term is represented by $\varepsilon_{it}$.

**Mediation effects model.** The following test model is based on the research methods of Wen et al. [59].

$$RD_{it}/Patent_{it} = \delta_0 + \delta_1 ESG_{it} + \delta_2 Controls_{it} + City_i + Year_t + \varepsilon_{it} \tag{3}$$

$$RDPeople_{it}/Sub_{it} = \gamma_0 + \gamma_1 ESG_{it} + \gamma_2 Controls_{it} + City_i + Year_t + \varepsilon_{it} \tag{4}$$

$$RD_{it}/Patent_{it} = \eta_0 + \eta_1 ESG_{it} + \eta_2 RDPeople_{it}/Sub_{it} + \eta_3 Controls_{it} + City_i + Year_t + \varepsilon_{it} \tag{5}$$

Where: R&D personnel (RDPeople) and government subsidies (Sub) are mediating variables.

Procedure for testing mediation effects: In the first step, the coefficient δ1 indicates the total effect of firms' ESG performance (ESG) on firms' innovation. If δ1 is significant, the test of equation (4) is continued; otherwise the test is stopped. In the second step, the coefficient γ1 measures whether there is a mediating effect of R&D people (RDPeople) and government subsidies (Sub); if γ1 is significant, it indicates a mediating effect. In the third step, we observe whether the coefficients η1 and η2 are significant.

In the empirical process, we used the Sgmediation command in the Stata 17.0 software to test for the significance of the mediation effect, including the Sobel test, Goodman test 1, and Goodman test 2, which the software automatically provided. In addition, the Bootstrap method was used to test for indirect effects.

## Indicator selection

**Explained variable: Enterprise innovation.** This study examines both innovation inputs and green innovation outputs for business innovation.

Innovation investment is more determined by business managers, and usually domestic and international studies define R&D investment as the ratio of R&D expenditures to sales revenues in the current year [60,61]. In this study, referring to Shen Yi's study [62], the absolute value index of R&D investment can more completely describe the R&D behavior of enterprises. In this study, corporate innovation investment (RD) is measured by the natural logarithm of the amount of R&D investment.

For the measurement of green innovation output of enterprises, the number of patent applications of enterprises is generally used [63], and the output value of new products of enterprises is also used [64]. Firstly, patent data can reflect

a more specific output of the invention process [65]. In addition, compared with China's information environment, patent application indicators reflect companies' innovation research data more. Therefore, this study refers to related studies to measure green innovation output by the number of enterprises' green patent applications (including green inventions and green utility models) [66].

**Explanatory variable: ESG performance.** The current evaluation of Chinese companies by mainstream international ESG rating agencies such as MSCI, Thomson Reuters, FTSE, etc., is not entirely in line with China's national conditions. For example, 'specialized poverty alleviation' is an important part of all enterprises' fulfillment of their social responsibility. However, the strength and results of corporate poverty alleviation have not been included in the ESG evaluation systems of international rating agencies. Therefore, based on the appropriateness and availability of ESG ratings, this paper adopts the CSI ESG ratings set for Chinese companies to reflect the ESG performance of the sample companies. It divides the CSI ESG ratings into nine grades ranging from AAA to C, assigning values from 9 to 1, respectively [67,68]. The related score increases with increasing ESG performance of a corporation.

**Mediating variable: R&D personnel and government subsidies.** The natural logarithm of the number of research and development (R&D) personnel was employed to measure the mediating variable, namely R&D personnel (RDPeople), which exerts an influence on firms from within.

The natural logarithm of the quantity of government subsidies (Sub) was utilized to measure the mediating variable, government subsidies (Sub), which affects companies from an external perspective [69].

**Control variable.** In light of previous studies [70–72], the following control variables were selected for analysis: profitability (ROA), financial leverage (LEV), corporate growth (DEV), second job alignment (DUA), percentage of independent directors (PID), and firm value (TQ). Please refer to Table 1 for a detailed description of the main variables employed in this study.

## Empirical analysis

### Descriptive statistical analyses

Table 2 shows the description of the ESG scores of companies from 2015 to 2021. From 2015 to 2022, the study's sample size expanded year by year, increasing from 142 to 256, with an average annual growth rate of 17%, and the total number of observations reached 1,635. The standard deviation (SD) decreased from 1.162 in 2021 to 0.842, representing a 27% reduction. The increase in the minimum value and the 25th percentile (p25), this indicates that the gap in ESG performance among enterprises has narrowed. The minimum ESG score remained one from 2015 to 2021, and rose to 2 in 2022, showing an improvement in the lowest level of ESG performance. The maximum ESG score was 6 in 2015–2017, 2019, and 2021, while it reached 7 in 2018, 2020, and 2022. This reflects that some companies have achieved outstanding ESG performance, and there is a growing trend in the number of companies with high ESG scores over time.

From 2015 to 2021, the p25 remained stable at 3, while the p75 mainly was 5, indicating that 25% of companies had ESG scores ≤3, and 75% had scores ≤5. In 2018, a local anomaly (p75=4, lower than the five observed in other years) reflected a generally weaker ESG performance across the industry that year. This may be related to events such as the impact of the pharmaceutical industry's centralized procurement policy and the capital winter, leading companies to neglect ESG considerations temporarily. However, p75 returned to 5 after 2019, indicating that the industry quickly recovered, demonstrating strong ESG resilience. Additionally, in 2022, p25 rose to 4 and p75 remained at 5, indicating an upward shift in ESG score distribution, a reduction in the proportion of companies in the lower score segment, and an optimized structure of industry ESG performance. The sudden change in 2022 may be attributed to both the impact of the dual carbon targets and stricter ESG regulations, and increased investor focus on ESG, which has compelled companies to improve their performance.

**Table 1. Related variable definitions.**

| Type | Name | Symbols | Definition |
|---|---|---|---|
| Explained variable | Innovative Inputs | RD | Natural logarithm of the amount invested in R&D |
| | Green Innovation Outputs | Patent | Number of green patent applications plus the natural logarithm of 1 (including green invention patents and green utility model patents) |
| Explanatory variable | ESG Performance | ESG | According to CSI ESG rating from low to high, the value is 1~9. |
| Mediating variable | R&D Personnel | RDPeople | Natural logarithm of the number of R&D staff |
| | Government Subsidy | Sub | Natural logarithm of government grants |
| Control variable | profitability | ROA | Net profit/total assets |
| | Financial Leverage | LEV | Total liabilities/total assets |
| | Enterprise Growth | DEV | (Current operating income – prior period's operating income)/ prior period's operating income |
| | Two Jobs in One | DUA | Chairman and general manager is the same person to take the value of 1, otherwise 0 |
| | Percentage of Independent Directors | PID | Percentage of Independent Directors/Directors |
| | Enterprise Value | TQ | Tobin's Q, total market capitalization at the end of the period/total assets at the end of the period |
| | Combined Tax Rate | CTR | (Business taxes and surcharges + income tax expense)/total profit |
| | cash flow ratio | CASH | Net cash flow/total funds |
| | Total Asset Turnover | ATO | Operating income/average total assets |
| | Time Effect | Year | Year dummy variables, assigned a value of 1 if it belongs to the year, otherwise assigned a value of 0 |
| | Regional Effect | CityCode | Region dummy variables, assigned a value of 1 if it belongs to the region, otherwise assigned a value of 0. |
| | Nature of Property Rights | Soe | State-owned enterprises take the value of 1, others take the value of 0. |
| | Nature of Area | Area | If the company's place of registration belongs to the eastern region takes the value of 1, belonging to the central and western regions takes the value of 0. |

**Table 2. Descriptive statistics of ESG scores.**

| Year | N | Mean | SD | Min | p25 | p50 | p75 | Max |
|---|---|---|---|---|---|---|---|---|
| 2015 | 142 | 3.873 | 1.030 | 1 | 3 | 4 | 5 | 6 |
| 2016 | 152 | 3.862 | 1.061 | 1 | 3 | 4 | 5 | 6 |
| 2017 | 189 | 3.862 | 1.017 | 1 | 3 | 4 | 5 | 6 |
| 2018 | 200 | 3.805 | 1.069 | 1 | 3 | 4 | 4 | 7 |
| 2019 | 209 | 3.789 | 1.249 | 1 | 3 | 4 | 5 | 6 |
| 2020 | 239 | 3.870 | 1.252 | 1 | 3 | 4 | 5 | 7 |
| 2021 | 248 | 3.851 | 1.162 | 1 | 3 | 4 | 5 | 6 |
| 2022 | 256 | 4.387 | 0.842 | 2 | 4 | 4 | 5 | 7 |
| Total | 1635 | 3.928 | 1.111 | 1 | 3 | 4 | 5 | 7 |

Table 3 shows the descriptive statistics results for every variable. Corporate innovation input, or RD, has a mean value of 18.323; its most significant value is 22.571, and its lowest value is 13.740. The degree of data dispersion is relatively small. The average corporate green innovation output (Patent) is 0.078, with a maximum of 2.079 and a minimum of 0. When combined with the standard deviation, the overall green innovation output of Chinese pharmaceutical listed companies is relatively low. With a mean value of 3.928, corporate ESG performance (ESG) shows that Chinese pharmaceutical businesses have an average level of ESG performance within the CCC-B grade range. This indicates that significant

**Table 3. Descriptive statistics(n = 1635).**

| Variable | Mean | p50 | SD | Min | Max |
|---|---|---|---|---|---|
| RD | 18.323 | 18.298 | 1.222 | 13.740 | 22.571 |
| Patent | 0.078 | 0 | 0.278 | 0 | 2.079 |
| ESG | 3.928 | 4.000 | 1.111 | 1.000 | 7.000 |
| ROA | 0.061 | 0.058 | 0.085 | −0.847 | 0.604 |
| LEV | 0.299 | 0.274 | 0.165 | 0.014 | 1.333 |
| DEV | 0.455 | 0.110 | 7.799 | −0.859 | 263.271 |
| DUA | 0.358 | 0 | 0.479 | 0 | 1.000 |
| PID | 0.373 | 0.333 | 0.050 | 0.300 | 0.600 |
| CTR | 0.222 | 0.212 | 0.933 | −21.656 | 9.875 |
| CASH | 0.073 | 0.068 | 0.072 | −0.260 | 0.726 |
| ATO | 0.506 | 0.482 | 0.226 | 0.004 | 2.420 |
| TQ | 2.611 | 1.986 | 1.912 | 0.715 | 22.572 |
| Sub | 16.597 | 16.634 | 1.410 | 0 | 20.127 |
| RDPeople | 5.450 | 5.460 | 0.956 | 1.792 | 8.608 |

potential for enhancement remains. The mean value of research and development personnel (RDPeople) is 5.450, with a maximum value of 8.608 and a minimum of 1.792, which signifies the existence of deficiencies in the quantity of research and development personnel in Chinese pharmaceutical listed companies. The average value of government subsidies (Sub) is 16.597, with a high of 20.127 and a low of 0, which indicates a significant disparity in the government subsidies obtained by Chinese pharmaceutical listed companies. It is evident that pharmaceutical companies still have significant opportunities for advancement in innovation, particularly in green innovation, which would enable them to signal to the government their willingness to engage in environmentally-friendly practices, thereby influencing the allocation of government resources.

## Correlation analysis

According to the correlation coefficient between the main variables (Table 4). At the 1% level, there is a substantial and positive correlation between the key explanatory variables (ESG), corporate innovation (RD), and patents of Chinese pharmaceutical listed businesses. This finding provides preliminary support for hypothesis H1. Government subsidies and R&D staff have correlation values of 0.205 and 0.229 concerning ESG performance, respectively. At the 1% level, both coefficients exhibit a substantial positive trend. It can be preliminarily confirmed that the positive ESG performance of pharmaceutical listed companies is conducive to increasing the intensity of enterprises' investment in R&D personnel and is favorable to enterprises' access to government resources and more government subsidies. Due to several control variables, additional tests for multicollinearity are necessary, as the correlation analysis can only quantify the correlation

**Table 4. Correlation analysis of major variables.**

| | RD | Patent | ESG | RDPeople | Sub |
|---|---|---|---|---|---|
| RD | 1 | | | | |
| Patent | 0.093*** | 1 | | | |
| ESG | 0.257*** | 0.106*** | 1 | | |
| RDPeople | 0.798*** | 0.135*** | 0.229*** | 1 | |
| Sub | 0.689*** | 0.106*** | 0.205*** | 0.613*** | 1 |

***, **, and *, respectively, denote significance at the 1%, 5%, and 10% levels.

between the two variables. The test results show that the VIF values are all below 5, suggesting that the variables are highly non-correlated and will not impact the primary regression analysis.

## Regression analysis

Table 5 reports benchmark regressions of pharmaceutical firms' ESG performance on firm innovation. Columns (1) and (4) present results controlling for region effects. Columns (2) and (5) are regressions controlling for time effects. Columns (3) and (6) are regressions controlling for both area and time effects. In columns (3) and (6), it is shown that the coefficients of (ESG) are divided into 0.220 and 0.021, which are both significant at the 1% level according to the statistical analysis. The ESG performance of pharmaceutical businesses positively influences both corporate and green innovation's input and output. This result indicates that the superior ESG performance of pharmaceutical enterprises is conducive to the enterprises in question improving their innovation input and promoting green innovation output. As a result, the H1a and H1b hypotheses have been confirmed. Based on the findings of the tests indicated above, it can be concluded that pharmaceutical companies' superior ESG performance has a favorable impact on the advancement of corporate innovation.

## Robustness tests

**Replacement of explained variables.** Utilizing the natural logarithm of a company's R&D expenditure neglects the significant contribution of R&D staff to innovation. This result may be attributed to the inherent unpredictability associated with the impact of R&D personnel inputs on firms [73]. This study uses the natural logarithm of businesses' R&D investment and the natural logarithm of the number of R&D workers to construct an interaction term (RD1) in the robustness test [74]. This study remeasures firms' green innovation output by calculating the natural logarithm of the number of green patents obtained during the current period plus one.

In addition to this, it is essential to consider the possibility of reverse causality as well as the lag effect that the ESG performance of companies has on their innovation. Good ESG performance in the early period provides a solid foundation for firms' development and innovation in the later period. Therefore, regression with explanatory variables lagged by one period (L.ESG) is a helpful approach. The findings are displayed in columns (1) and (2) of Table 6, demonstrating that the coefficient for the variable (L.ESG) is significantly positive, which suggests that the previous findings are robust.

**Table 5. Baseline regression results.**

| VARIABLES | (1) | (2) | (3) | (4) | (5) | (6) |
|---|---|---|---|---|---|---|
| | RD | RD | RD | Patent | Patent | Patent |
| ESG | 0.248*** | 0.279*** | 0.220*** | 0.016** | 0.030*** | 0.021*** |
| | (9.47) | (10.25) | (8.43) | (2.22) | (4.29) | (2.71) |
| Constant | 16.892*** | 17.056*** | 17.000*** | 0.033 | −0.013 | 0.012 |
| | (68.99) | (74.68) | (69.67) | (0.52) | (−0.24) | (0.18) |
| Control variables | YES | YES | YES | YES | YES | YES |
| Observations | 1,632 | 1,635 | 1,632 | 1,632 | 1,635 | 1,632 |
| R-squared | 0.544 | 0.219 | 0.586 | 0.145 | 0.034 | 0.154 |
| CityCode FE | YES | | YES | YES | | YES |
| Year FE | | YES | YES | | YES | YES |
| F test | 0 | 0 | 0 | 0.0372 | 2.57e-05 | 0.0166 |
| F | 30.91 | 37.72 | 28.13 | 1.932 | 3.934 | 2.181 |

Robust t-statistics in parentheses, *** $p<0.01$, ** $p<0.05$, * $p<0.1$.

**Table 6. Results of robustness tests.**

| VARIABLES | (1) | (2) | (3) | (4) | (5) | (6) |
|---|---|---|---|---|---|---|
| | RD 1 | Patent 1 | RD(q30) | RD(q60) | RD(q90) | Patent 2 |
| ESG | | | 0.194*** | 0.312*** | 0.408*** | 0.236** |
| | | | (5.82) | (7.96) | (9.82) | (2.33) |
| L.ESG | 3.955*** | 0.014** | | | | |
| | (7.34) | (1.97) | | | | |
| Constant | 77.405*** | −0.053 | 16.301*** | 17.064*** | 18.526*** | −4.582*** |
| | (14.93) | (−0.80) | (57.53) | (49.51) | (42.56) | (−4.49) |
| Control variables | YES | YES | YES | YES | YES | YES |
| Observations | 1,372 | 1,372 | 1,635 | 1,635 | 1,635 | 1,635 |
| R-squared | 0.556 | 0.212 | | | | |
| CityCode FE | YES | YES | | | | YES |
| Year FE | YES | YES | | | | YES |
| F test | 0 | 0.00236 | | | | 0 |
| F | 26.30 | 2.748 | | | | . |

Robust t-statistics in parentheses, *** p<0.01, ** p<0.05, * p<0.1.

**Replacement models.** Outliers exert a greater influence on the estimation outcomes in mean reversion. In contrast, quantile regression is founded upon the quartiles of the data for estimation, thereby conferring greater resilience to the impact of outliers and ensuring more robust estimation outcomes. This study utilizes quantile regression to evaluate company ESG performance and innovation investment robustness. Columns (3), (4), and (5) in Table 6 present the outcomes of various regression models. A substantial positive connection exists between enterprises' ESG performance and innovation investment, with the correlation coefficient progressively increasing as the quantile of innovation investment rises, indicating the robustness of the findings.

Patent applications are non-negative integers, count variables and discrete variables applicable to Poisson regression models. This work re-evaluates the empirical test of innovation output robustness using the Poisson regression model. Table 6, column (6), displays the outcomes of the substitution regression model. The regression findings demonstrate that the coefficient estimate for the ESG variable is significantly optimistic at the 5% significance level. The strength of the benchmark regression is shown by the replacement model's mostly consistent regression findings.

## Endogeneity discussion

Because of the potential bidirectional causality that may exist between ESG performance and firm innovation, coupled with the possibility of omitted variable bias in model construction, the instrumental variables approach and systematic generalized method of moments estimation (GMM-SYS) are adopted to re-test the impact of ESG performance and firm innovation in order to mitigate the endogeneity effect due to the omitted variables and bidirectional causality, etc.

This study draws on Yang Y, Meng M et al. [75–77], uses the explanatory variable lagged one period (L.ESG) as an instrumental variable, and employs 2SLS for regression analysis. Firstly, this paper carried out the test of non-identifiability, and the analysis of the KP-LM statistic showed significant results, which strongly rejected the original hypothesis of 'non-identifiability'. It confirmed the correlation between the instrumental and core explanatory variables, indicating that the instrumental variables selected in this paper are valid and reliable. Secondly, the KP-Wald F-statistic of 348.270 is much higher than the 10% critical value of 16.38, indicating no problem with weak instrumental variables. Therefore, the choice of instrumental variables is justified. No over-identification test is required since the number of instrumental variables equals the number of endogenous explanatory variables. Finally, the paper conducts the 2SLS test

for instrumental variables. In the first stage regression results, the regression coefficients of L.ESG are all significantly positive at the 1% level (Table 7), so L.ESG is related to firms' ESG performance, which meets the correlation assumption. In the second-stage regression results, the regression coefficients of ESG are significantly positive at the 1% and 5% levels (Table 7), consistent with the benchmark regression, indicating that the ESG performance of listed Chinese pharmaceutical companies can improve corporate innovation.

The GMM-SYS results are shown in Table 8, which may be found here. Equally, the coefficients of ESG performance are 0.104 and 0.245. The results are dependable since the findings show that both coefficients are positive at the 5% level. According to Hansen's test, the p-values were 0.171 and 0.195, respectively, which passed the overidentification test. While none of the second-order differences are autocorrelated, both p-values of AR (1) are less than 0.1, and both p-values of AR (2) are more than 0.1, indicating that there is autocorrelation in all the first-order differences of the perturbation terms. Because of this, the model is successful in passing the autocorrelation test. The conclusion is that the regression findings produced through the use of the GMM-SYS approach agree with the findings of the baseline regression.

**Table 7. Instrumental variables method.**

| VARIABLES | Stage One | Stage Two | Stage One | Stage Two |
| --- | --- | --- | --- | --- |
| | ESG | RD | ESG | Patent |
| L.ESG | 0.447*** (18.66) | | 0.447*** (18.66) | |
| ESG | | 0.470***(8.77) | | 0.0352**(2.07) |
| KP-LM Statistic [P] | | 298.778*** (0.00) | | 298.778*** (0.00) |
| KP-Wald F Statistic [P] | | 348.270 | | 348.270 |
| 10% maximal IV size | | 16.38 | | 16.38 |
| N | 1372 | 1372 | 1372 | 1372 |

t statistics in parentheses, * p<0.1, ** p<0.05, *** p<0.01.

**Table 8. System GMM method.**

| VARIABLES | (1) | (2) |
| --- | --- | --- |
| | RD | Patent |
| L.RD | 0.973***(0.029) | |
| ESG | 0.104**(0.049) | 0.245**(0.119) |
| L.Patent | | −0.519***(0.076) |
| Constant | 0.749 | −1.090 |
| | (0.665) | (1.108) |
| Control variables | YES | YES |
| Number of id | 248 | 248 |
| AR (1) | −5.940 | −2.967 |
| AR (1)p | 2.85e-09 | 0.00301 |
| AR (2) | −0.414 | −1.465 |
| AR (2)p | 0.679 | 0.143 |
| Hansen | 45.04 | 36.42 |
| Hansenp | 0.171 | 0.195 |
| N | 1375 | 1375 |

Standard errors in parentheses, *** p<0.01, ** p<0.05, * p<0.1.

## Heterogeneity analysis

**Heterogeneity of enterprises' property attributes.** This study further differentiates Chinese pharmaceutical listed companies into state-owned and non-state-owned enterprises and examines the influence of corporate ESG performance on the innovation of these two types of companies. Table 9 shows the outcomes. Consult columns (1) and (2) for specifics of the innovative sources. At the 1% level, the regression coefficients for the variable (ESG) for both SOEs and non-SOEs are positive. In particular, it is found that state-owned enterprises (SOEs) have a smaller regression coefficient of the variable than non-state-owned businesses. The results for green innovation output are presented in columns (3) and (4). The results for the ESG performance of SOEs (ESG) are insignificant; however, the regression coefficients for the non-SOE variable (ESG) are considerably positive (5%). On the one hand, SOEs are supposed to satisfy a broader spectrum of social obligations and responsibilities. On the other hand, they are typically not subject to the same degree of performance pressures as their private sector counterparts, and therefore do not have to engage in the same level of strategic thinking when it comes to maximizing corporate profits. The superior ESG performance of non-state-owned firms is conducive to greater innovation in the pharmaceutical industry than that observed in state-owned firms.

**Heterogeneity of enterprises regional.** There are notable discrepancies in economic advancement across China's eastern, central, and western regions. Consequently, the consequences of firms' enhanced ESG practices in disparate regions will vary. Table 10 summarizes the implications of geographical heterogeneity on company ESG performance and innovation. The calculated coefficients on the variable for Eastern region firms (ESG) are significant at the level of one percent for both company innovation inputs and green innovation outputs. Though the coefficients are less than in the Eastern region, the variable for companies in the Central and Western regions (ESG) is notably positive at the 1% level for innovation inputs. The variable for firms in the Midwest (ESG) is not statistically significant for innovation output. The influence of ESG performance (ESG) factors significantly enhances innovation in pharmaceutical companies located in the eastern area. However, this effect is less pronounced for those in the central and western regions. Consequently, pharmaceutical companies located in the eastern region are more adept at strengthening their ESG performance to drive corporate innovation, especially during green innovation output.

**Heterogeneity of enterprise size.** Pharmaceutical companies engage in more risky innovation activities and have longer payoff cycles. Larger firms are typically better financed than smaller firms and are therefore able to leverage the resources that come with improved ESG performance to drive innovation in their organizations. This research groups

**Table 9. Results of heterogeneity test for property attributes.**

| VARIABLES | (1) | (2) | (3) | (4) |
|---|---|---|---|---|
| | non-SOEs | SOEs | non-SOEs | SOEs |
| | RD | RD | Patent | Patent |
| ESG | 0.195*** | 0.181*** | 0.019** | 0.010 |
| | [6.46] | [3.97] | [2.19] | [0.51] |
| Constant | 17.203*** | 16.852*** | 0.031 | 0.068 |
| | [56.35] | [49.04] | [0.40] | [0.43] |
| Control variables | YES | YES | YES | YES |
| N | 1297 | 331 | 1297 | 331 |
| CityCode FE | YES | YES | YES | YES |
| Year FE | YES | YES | YES | YES |
| r2_a | 0.524 | 0.788 | 0.072 | 0.171 |
| F | 17.368 | 11.212 | 1.871 | 0.648 |

t statistics in brackets, * p<0.1, ** p<0.05, *** p<0.01.

**Table 10. Results of the regional heterogeneity test.**

| VARIABLES | (1) | (2) | (3) | (4) |
|---|---|---|---|---|
| | Midwest | East | Midwest | East |
| | RD | RD | Patent | Patent |
| ESG | 0.178*** | 0.234*** | 0.013 | 0.025*** |
| | [4.64] | [6.91] | [0.89] | [2.72] |
| Constant | 16.519*** | 17.259*** | 0.014 | 0.006 |
| | [45.88] | [53.71] | [0.13] | [0.07] |
| Control variables | YES | YES | YES | YES |
| N | 678 | 954 | 678 | 954 |
| CityCode FE | YES | YES | YES | YES |
| Year FE | YES | YES | YES | YES |
| r2_a | 0.593 | 0.485 | 0.083 | 0.084 |
| F | 9.536 | 27.870 | 1.523 | 1.602 |

t statistics in brackets, * $p < 0.1$, ** $p < 0.05$, *** $p < 0.01$.

companies based on their size, separating small and medium-sized businesses (SMEs) from big companies (LEs). The revenue of each firm is then compared to the 70th percentile to determine whether it exceeds this threshold. This allows for the testing of two different sizes of sample. Table 11 shows how firm size variation influences firm innovation and ESG performance. The computed coefficients of ESG performance in innovation investment show that all are significant at the 1% significance level. However, the coefficients of ESG performance for LEs are larger than those for SMEs. This result suggests that the ESG performance of larger pharmaceutical companies is associated with a greater propensity to invest in innovation. Regarding green innovation output, the ESG performance of large enterprises is not statistically significant.

In contrast, the ESG performance of small and medium-sized enterprises has a substantial coefficient at the 5% level. This result suggests that smaller pharmaceutical listed businesses' ESG performance is better at encouraging the output of green innovation. This result can be ascribed to the fact that pharmaceutical LEs have higher R&D technology

**Table 11. Results of enterprise size heterogeneity test.**

| VARIABLES | (1) | (2) | (3) | (4) |
|---|---|---|---|---|
| | SMEs | LEs | SMEs | LEs |
| | RD | RD | Patent | Patent |
| ESG | 0.097*** | 0.168*** | 0.015* | 0.002 |
| | [3.66] | [5.06] | [1.76] | [0.12] |
| Constant | 17.356*** | 19.158*** | 0.060 | −0.101 |
| | [65.28] | [57.18] | [0.82] | [-0.62] |
| Control variables | YES | YES | YES | YES |
| N | 1136 | 484 | 1136 | 484 |
| CityCode FE | YES | YES | YES | YES |
| Year FE | YES | YES | YES | YES |
| r2_a | 0.499 | 0.697 | 0.095 | 0.186 |
| F | 3.410 | 7.622 | 1.563 | 1.253 |

t statistics in brackets, * $p < 0.1$, ** $p < 0.05$, *** $p < 0.01$.

requirements and longer R&D cycles than SMEs, resulting in a time lag in green innovation output, which is not as pronounced as that of SMEs in the short term.

## Mediation effect test

**Mediation effect test based on R&D personnel.** Columns (1) to (3) of Table 12 present the outcomes of assessing firms' ESG performance compared to their innovation inputs. Columns (4) to (6) of Table 12 present the results of testing firms' ESG performance against firms' innovation output. The mediation effect test's first steps, shown in columns (1) and (4), respectively, examine the role of the explanatory variable firm ESG performance on the variables firm innovation input (RD) and firm innovation output (Patent). The variable (ESG) coefficients are positive, demonstrating that improved ESG performance contributes to increased corporate innovation. This result supports the hypothesis H1. Columns (2) and (5) represent the second step of the mediation effect test, which examines the influence of the explanatory variable, firm ESG performance, on the variable firm investment in R&D staff. The coefficients for the variable (ESG) are all statistically significant at the 1% level, indicating that superior ESG performance enhances the enterprise's capacity to augment its investment in R&D staff (RDPeople). This result supports the hypotheses H2a and H2b. Columns (3) and (6) represent the mediation effect test step three. According to the fact that the coefficient of the variable (RDPeople) is significantly positive, it can be deduced that the contribution of R&D staff is a significant component that influences the innovation of businesses. The coefficients of ESG performance of firms (ESG) on innovation input (RD) and innovation output (Patent), although still significant, are significantly lower compared with columns (1) and (4). This result suggests a partial mediation influence of R&D personnel (RDPeople) on how innovation and a company's ESG performance are related.

**Table 12. Mediation effects test based on R&D personnel.**

| VARIABLES | (1) | (2) | (3) | (4) | (5) | (6) |
|---|---|---|---|---|---|---|
| | RD | RDPeople | RD | Patent | RDPeople | Patent |
| ESG | 0.220*** | 0.167*** | 0.077*** | 0.021*** | 0.167*** | 0.016** |
| | (8.43) | (8.18) | (4.37) | (2.71) | (8.18) | (2.10) |
| RDPeople | | | 0.858*** | | | 0.028** |
| | | | (32.24) | | | (2.40) |
| Constant | 17.000*** | 4.572*** | 13.075*** | 0.012 | 4.572*** | −0.117 |
| | (69.67) | (22.73) | (66.94) | (0.18) | (22.73) | (−1.36) |
| Control variables | YES | YES | YES | YES | YES | YES |
| Observations | 1,632 | 1,632 | 1,632 | 1,632 | 1,632 | 1,632 |
| R-squared | 0.586 | 0.535 | 0.796 | 0.154 | 0.535 | 0.159 |
| F test | 0 | 0 | 0 | 0.0166 | 0 | 0.0154 |
| r2_a | 0.551 | 0.497 | 0.779 | 0.0842 | 0.497 | 0.0887 |
| F | 28.13 | 29.32 | 160.4 | 2.181 | 29.32 | 2.139 |
| Sobel Test | 0.1433*** (z=8.635) | | | 0.0047*** (z=2.764) | | |
| Goodman Test1 | 0.1433*** (z=8.633) | | | 0.0047*** (z=2.748) | | |
| Goodman Test2 | 0.1433*** (z=8.638) | | | 0.0047*** (z=2.780) | | |
| Indirect effect | 0.1433*** (z=8.635) | | | 0.0047*** (z=2.764) | | |
| Direct effect | 0.0766*** (z=4.685) | | | 0.0162** (z=2. 237) | | |
| Total effect | 0.2199*** (z=9.680) | | | 0.0209*** (z=2. 951) | | |
| Indirect effect to total effect ratio | 0.6518 | | | 0. 2243 | | |
| Bootstrap Method Indirect Effects Test | 0.1433*** (LLCI=0.1403, ULCI=0.1439) | | | 0.0047*** (LLCI=0.0037, ULCI=0.0059) | | |
| | Effective mechanisms-Positive transfer | | | Effective mechanisms-Positive transfer | | |

Robust t-statistics in parentheses, *** p<0.01, ** p<0.05, * p<0.1.

The aforementioned results validate the relationship between corporate ESG performance and corporate innovation, specifically "ESG performance - R&D personnel - corporate innovation". The mediating influence of R&D professionals constitutes 65.18% of innovation input, accounting for 22.43% of green innovation output. This paper posits that the stepwise regression method for testing mediation effects is less reliable. Consequently, the Sgmediation command test and the self-sampling method (Bootstrap) were employed to investigate this hypothesis further. The Sgmediation command test conducted provided three tests of significance, Sobel, Goodman1, and Goodman2, all found to be significant. The self-sampling standard error and self-sampling confidence intervals were calculated, and the confidence intervals of R&D personnel on the indirect effects of innovation investment and ESG performance did not include 0 (LLCI = 0.1403, ULCI = 0.1439). This result indicates a significant indirect effect, which is 0.1433. The confidence interval of R&D personnel on the indirect effect of green innovation output and ESG performance also does not include 0, and the indirect effect is significant, with an indirect effect of 0.0047. The results support hypothesis H2.

**Mediation effect test based on government subsidies.** Table 13 presents the findings of the mediation effect analysis for the variable government subsidy (Sub). The test procedure for columns (1) and (4) is identical to that employed in Table 12 and will not be repeated here. The ESG performance of companies positively impacts their access to government subsidies, as evidenced by the strongly positive coefficients of the ESG variable in columns (2) and (5) of Step 2 of the mediation effect test. Therefore, this preliminary test of H3a and H3b is valid. The coefficients of variable (Sub) in columns (3) and (6), as outlined in step 3 of the mediation effect test, are both significantly positive, indicating that firms receiving greater government subsidies facilitate the advancement of corporate innovation. In addition, the variable (ESG) is significant in both columns (3) and (6), but the value of the coefficient decreases compared to columns (1) and (4). The results suggest that government subsidies partially influence the relationship between corporate ESG performance and innovation. The "ESG performance – government subsidy – firm innovation" mechanism has been validated, thereby supporting hypothesis H3. It can be observed that government subsidies mediate 53.44% of innovation inputs and 21.95% of green innovation outputs.

Furthermore, the three significance tests of Sobel, Goodman1 and Goodman2 were met. The results of calculating the self-sampling standard error and self-sampling confidence intervals revealed that the indirect effect's confidence interval did not contain zero and was significant. The indirect effect was 0.1175 and 0.0046. These findings supported hypothesis H3.

## Conclusions and recommendations

### Conclusions

Using unbalanced panel data of A-share pharmaceutical listed companies from 2015 to 2022, this study examines the impact of ESG performance on corporate innovation and development of Chinese pharmaceutical listed companies, as well as provides new insights into the mechanism of the role between ESG performance and corporate innovation of pharmaceutical firms, which broadens the research horizons of exploring the path of innovation of pharmaceutical firms, and enriches the research content of the impact of ESG practices on the long-term sustainable development of pharmaceutical companies. The results of the study are as follows.

(1)  There is a significantly positive connection between a company's environmental, social, and governance (ESG) performance and the company's innovativeness. Once several robustness tests have been completed and potential endogeneity problems have been resolved, the previously presented findings remain consistent.

(2) The heterogeneity analysis indicated that the ESG performance of pharmaceutical manufacturing firms significantly impacts innovation in non-state-owned organizations and those located in the eastern area. In contrast, the ESG performance of major corporations significantly affects innovation inputs, whereas the ESG performance of small firms has a greater impact on innovation outputs.

**Table 13. Mediation effect tests based on government subsidies.**

| VARIABLES | (1) | (2) | (3) | (4) | (5) | (6) |
|---|---|---|---|---|---|---|
| | RD | Sub | RD | Patent | Sub | Patent |
| ESG | 0.220*** | 0.224*** | 0.102*** | 0.021*** | 0.224*** | 0.016** |
| | (8.43) | (8.00) | (4.97) | (2.71) | (8.00) | (2.10) |
| Sub | | | 0.524*** | | | 0.020*** |
| | | | (23.49) | | | (2.64) |
| Constant | 17.000*** | 15.731*** | 8.755*** | 0.012 | 15.731*** | −0.310** |
| | (69.67) | (59.00) | (21.99) | (0.18) | (59.00) | (−2.31) |
| Control variables | YES | YES | YES | YES | YES | YES |
| Observations | 1,632 | 1,632 | 1,632 | 1,632 | 1,632 | 1,632 |
| R-squared | 0.586 | 0.523 | 0.730 | 0.154 | 0.523 | 0.158 |
| F test | 0 | 0 | 0 | 0.0166 | 0 | 0.0120 |
| r2_a | 0.551 | 0.484 | 0.708 | 0.0842 | 0.484 | 0.0886 |
| F | 28.13 | 35.13 | 90.14 | 2.181 | 35.13 | 2.210 |
| Sobel Test | 0.1175*** (z = 8.367) | | | 0.0046*** (z = 2.735) | | |
| Goodman Test1 | 0.1175*** (z = 8.363) | | | 0.0046*** (z = 2.719) | | |
| Goodman Test2 | 0.1175*** (z = 8.372) | | | 0.0046*** (z = 2.751) | | |
| Indirect effect | 0.1175*** (z = 8.367) | | | 0.0046*** (z = 2.735) | | |
| Direct effect | 0.1024*** (z = 5.447) | | | 0.0163** (z = 2.252) | | |
| Total effect | 0.2199*** (z = 9.680) | | | 0.0209*** (z = 2.951) | | |
| Indirect effect to total effect ratio | 0. 5344 | | | 0. 2195 | | |
| Bootstrap Method Indirect Effects Test | 0.1175*** (LLCI = 0.1076, ULCI = 0.1435) | | | 0.0046*** (LLCI = 0.0029, ULCI = 0.0058) | | |
| | Effective mechanisms-Positive transfer | | | Effective mechanisms-Positive transfer | | |

Robust t-statistics in parentheses, *** p < 0.01, ** p < 0.05, * p < 0.1.

(3) Further research on the mechanism of action indicates that good ESG performance can facilitate enterprise innovation through external government subsidies and internal R&D personnel input. The correlation between ESG performance and corporate innovation in the pharmaceutical manufacturing sector is mediated by government subsidies and input from R&D personnel.

## Insights and recommendations

Policy implications:

(1) Establishment of differentiated ESG incentive policies. For enterprises with poor ESG performance but significant improvement, the Government can guide social capital to set up special incentive funds (e.g., green funds, ESG investment funds) to provide innovation financial support and technical support to enterprises with weak R&D capabilities and pharmaceutical enterprises in central and western China, and to help them enhance their innovation capabilities and levels. Provide more policy support and incentives to enterprises with excellent ESG performance. For example, tax incentives, green credits and low-interest loans can be used to encourage enterprises to invest more in corporate environmental, social and corporate governance.

(2) Improve the mechanism for training R&D talents. The Government should cooperate with enterprises to improve the mechanism for training, research, and development personnel, especially in green technology and sustainable development. Setting up special funds, providing training opportunities, and supporting school-enterprise cooperation can nurture more ESG-aware R&D talents.

(3) To optimize the allocation of government subsidies and focus on supporting green innovation by companies with strong ESG performance, the government should formulate a clear policy on the scope, conditions and use of subsidies to ensure that the subsidy funds are used effectively to promote the innovative activities of pharmaceutical companies.

Business Side Recommendations:

(1) Enhancing environmental awareness and practices is paramount. Chinese pharmaceutical businesses listed on the stock exchange must fully recognize the pivotal role of ESG performance in fostering corporate innovation. They should focus on ESG principles, implement clean and low-carbon production methods, embrace social responsibility, enhance governance, and amass cash for R&D investment to attain sustainable development.

(2) Reinforce corporate governance. Focusing on the attraction and development of innovative talent, the hiring of more R&D staff for practical exploration, the enhancement of corporate R&D capabilities and the empowerment of corporate innovation will help maximize the favorable impact of corporate ESG performance on corporate innovation.

(3) Enhance transparency and information disclosure. Enhancing communication and interaction with stakeholders, ensuring the timely disclosure of information, and improving transparency and information disclosure are necessary. A favorable ESG rating is conducive to enhancing the enterprise's reputation, thereby facilitating the acquisition of strategic resources and empowering corporate innovation.

(4) Businesses are actively promoting cooperation and sharing. Enterprises are actively promoting cooperation and sharing, breaking down traditional barriers to competition and realizing efficient allocation of resources and complementary advantages by building an open and collaborative innovation ecosystem. For example, enterprises can fully exploit their respective advantages in technology, talent and market to form a synergy of collaborative innovation by building joint laboratories, sharing clinical trial data and cooperating in the development of new technology platforms.

## Limitations and prospects

The research in this paper still has some limitations.

First, although the CSI ESG scoring system is widely used in Chinese academia, it does not set index weights or assign specific indicators according to the characteristics of each industry. Follow-up research can improve the measurement method of corporate ESG performance.

Second, this study is based on pharmaceutical listed companies, however, there are differences in the sensitivity of pharmaceuticals to ESG elements in different segments of the manufacturing industry, such as biopharmaceuticals and traditional Chinese medicine, which leads to a lack of universality in the conclusions, which an in-depth study of the differences in the impact of ESG performance on corporate innovation in different segments can follow. In addition, for different industries, the impact of ESG performance on corporate innovation may vary. Follow-up studies can select other resource-sensitive, environment-sensitive, and special industries for comparative studies with the pharmaceutical industry.

Third, the time frame examined in this study was from 2015 to 2022, and data during the epidemic were not excluded. Although the volatility of the data due to the epidemic may affect the generality of the results, only three years of data from 2020 to 2022 are involved, so the findings are still informative. Future studies could consider excluding data from the epidemic period for re-analysis. In addition, given the rapid changes in the external environment, such as policies and economic conditions, present studies are time-sensitive. They may not fully reflect the latest trends in the future. It is recommended that consideration be given to using newer data or extending the time period of the study in future studies to validate the stability of the findings of this study.

## Supporting information

**S1 File.** **Table 4.** Correlation analysis of all variables. **Table 5.** Baseline regression results. **Table 6.** Results of robustness tests. **Table 8.** System GMM method. **Table 9**. Results of heterogeneity test for property attributes. **Table 10**. Results of the regional heterogeneity test. **Table 11**. Results of enterprise size heterogeneity test. **Table 12** Mediation effects test based on R&D personnel. **Table 13.** Mediation effect tests based on government subsidies.
(DOCX)

## Author contributions

**Conceptualization:** Liqiang Li, Su Wang, Yuwen Chen.

**Data curation:** Liqiang Li.

**Formal analysis:** Liqiang Li.

**Methodology:** Liqiang Li.

**Project administration:** Yuwen Chen.

**Supervision:** Su Wang, Yuwen Chen.

**Writing – original draft:** Liqiang Li.

**Writing – review & editing:** Su Wang, Yuwen Chen.

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
