## [Decision Letter · Decision Letter 0]

3 Jan 2025

Dear Dr. Chen,

Thank you for submitting your manuscript to PLOS ONE. After careful consideration, we feel that it has merit but does not fully meet PLOS ONE’s publication criteria as it currently stands. Therefore, we invite you to submit a revised version of the manuscript that addresses the points raised during the review process.

We look forward to receiving your revised manuscript.

Kind regards,

María del Carmen Valls Martínez, Ph.D.

Academic Editor

PLOS ONE

Comments from PLOS Editorial Office: We note that one or more reviewers has recommended that you cite specific previously published works. As always, we recommend that you please review and evaluate the requested works to determine whether they are relevant and should be cited. It is not a requirement to cite these works. We appreciate your attention to this request.

Reviewers' comments:

Reviewer's Responses to Questions

**Comments to the Author**

1. Is the manuscript technically sound, and do the data support the conclusions?

Reviewer #1: Partly

Reviewer #2: Yes

Reviewer #3: Yes

Reviewer #4: Yes

2. Has the statistical analysis been performed appropriately and rigorously?

Reviewer #1: No

Reviewer #2: No

Reviewer #3: N/A

Reviewer #4: Yes

3. Have the authors made all data underlying the findings in their manuscript fully available?

Reviewer #1: Yes

Reviewer #2: Yes

Reviewer #3: Yes

Reviewer #4: Yes

4. Is the manuscript presented in an intelligible fashion and written in standard English?

Reviewer #1: No

Reviewer #2: Yes

Reviewer #3: No

Reviewer #4: No

Reviewer #1: Summary: This study empirically investigates the effect of ESG performance on corporate innovation based on pertinent data of China's A share pharmaceutical listed businesses from 2015 to 2022, and investigates the influence channels of corporate R&D staff investment and government subsidies. It finds that better ESG performance boosts innovation and green innovation output, with stronger effects in non-state-owned firms and those in the eastern region. The study also suggests that R&D staff investment and government subsidies play significant roles in enhancing ESG performance and innovation.

Overall, the study is clear, but there are still some issues that need to be addressed by the authors.

a: Introduction

The section lacks an outline of the article's marginal contributions and an overview of each section.

b: Theoretical Basis and Research Hypothesis

b1: It is recommended that the background discussion should be extensive and the hypothesis development should be thoroughly examined through an extensive literature review. It is suggested to include references on government subsidies, such as: Han, Y., Han, L., Liu, C., & Wang, Q. (2024). How does government R&D subsidies affect enterprises’ viability? An investigation on inverted U-shaped relationship[J]. Finance Research Letters, 106235. https://doi.org/10.1016/j.frl.2024.106235

b2: I did not see an in-depth analysis of the Theoretical Basis. It is recommended to add Institutional background.

c. Research Design

c1: Please explain proprietary names such as ST, *ST, and PT to enable the reader to better understand their specific meaning, e.g. ST companies (Companies with special treatment due to two consecutive years of operating losses).

c2: The authors selected China's A-share pharmaceutical listed enterprises from 2015 to 2022 as the research sample. It would be beneficial if the authors could explain further and justify the strong reasons for only using pharmaceutical listed enterprises from 2015 to 2022. Is there any possibility of bias? If so, how will the authors address this issue?

d: Empirical Analysis

d1: The choice of instrumental variables in the endogeneity discussion is not appropriate, and the results are not robust enough.

d2: Although the article conducts mechanism examinations and heterogeneity analyses, these sections are too brief.

Other suggestions: 1. Check the article for errors, such as: Error! Reference source not found. 2. The clarity of the Figures is insufficient. 3. The language and format of the article require professional editing.

Good luck with the revision!

Reviewer #2: I would like to begin by congratulating the authors. The choice of theme is appropriate and current. The manuscript is very well written and current. However, there are important points to be analyzed.

1) Both ESG and enterprise innovation are latent variables, that is, it is not advisable to measure them by a single value. Thus, the appropriate statistical technique to understand the relationships between the variables and test the hypotheses would be through structural equation modeling and not regression. If a regression were to be used, it should be of the "Ordinal Regression" type, which would allow it to model the dependence of a polytomous ordinal response on a set of predictors, which can be factors or covariates. The Ordinal Regression design is based on the methodology of McCullagh (1980, 1998). This is because the ESG variable was collected in such a way as to be presented as an ordinary variable.

2) Saying that ESG can be measured by "According to CSI ESG rating from low to high, the value is 1~9" is a very simplistic approach. In many studies, ESG is made up of more than a dozen variables. It is much more advisable for ESG to be a latent variable than an observable variable.

3) Converting ordinal variables into numerical variables is not appropriate. It is a statistically incorrect conversion and can generate distorted or unrealistic values. I understand that in PLOSONE alone there have been more than 16 articles in the last two years using this information incorrectly (https://doi.org/10.1371/journal.pone.0310447;
https://doi.org/10.1371/journal.pone.0299184;
https://doi.org/10.1371/journal.pone.0307535;
https://doi.org/10.1371/journal.pone.0295548;
https://doi.org/10.1371/journal.pone.0313686;
https://doi.org/10.1371/journal.pone.0301701, just to name a few). As an reviewer, I cannot help but point out that the conversion of ordinal variables into numeric variables and the statistical technique (regression for latent variables) used are incorrect and cannot be done in this way. The result generated ends up not having statistical validity. The correct thing to do would be to collect the variables in quantitative format and use structural equation modeling.

As minor points: 1) it is not necessary to present the formulas used. It is understood. 2) I suggest using articles about esg that were published in plosone.

Reviewer #3: Comments on “The impact of ESG performance on corporate innovation—empirical evidence based on Chinese pharmaceutical listed companies”

Summary:

This paper empirically examines the impact of ESG performance on corporate innovation using a sample of A-share pharmaceutical listed companies from 2015 to 2022. The authors utilize the high-dimensional fixed effect model and mediation effects model to test the impact of ESG performance on corporate innovation. Further, the authors conduct endogeneity analysis and heterogeneity analysis. I have several general comments that might be useful for the authors to improve this paper.

1. The abstract should be improved. Your findings can be expressed more clearly in the abstract, which should also briefly mention the policy implications.

2. The study's aim and contribution should be expressed more strongly. The introduction needs to be more relevant to the discussion. The introduction's discussion and research contributions should be redesigned.

3. There needs to be more literature review and discussion of why this topic matters and how it fills a gap in the literature.

4. Control variables, such as PID (percentage of independent directors) and DUA (second job alignment), are typically used in corporate governance studies. They may need more relevance to the specific innovation outcomes being modeled.

5. Some control variables, such as ROA, TQ, and LEV, might be highly correlated with each other, which could affect the stability of coefficient estimates in regression models.

6. There are typo mistakes and format issues. Such as line 212: “innovation inpu” and line 216: Error! Reference source not found.

7. The language quality could be better, but it needs serious improvement.

Reviewer #4: This study empirically examines the influence of corporate ESG performance on innovative advancement using a sample of listed Chinese pharmaceutical companies. The results show that there is a significantly positive connection between a company's environmental, social, and governance (ESG) performance and its ability to innovate. These results are supported by various tests of robustness and endogeneity, etc.

I personally find the manuscript very interesting, current and relevant. The article appears well written and complete. However, before accepting it for publication, I suggest the following minor changes.

I suggest synthesizing the abstract, removing elements of detail, reviewing it following the IMRaD (Introduction-Methods-Results-Discussion) model, frequently used in empirical research in the natural and social sciences. In particular, the reference is to Methods M.

I find the introduction clear, as well as other parts of the paper. However, I would suggest presenting the objective now and better supporting the rest of the content. For example, some quotes could be added to support this: “The comprehensive advancement of China's manufacturing sector continues to demonstrate elevated energy consumption and significant pollution, accompanied by a low resource utilization rate, resulting in the degradation of the ecological environment.”.

Other suggestions for integration: formalize the research question and highlight the classic “Section” summary passage that closes introductory manuscripts.

(There are several passages in the document with the following text “Error! Reference source not found")

In “Research Design,” the number of companies analyzed should be better specified. It is suggested to first describe the variables used “RDit,” “Patentit,” and “ESGit,” arguing concisely why they are the ones used and useful for the research. Personal curiosity: why are these variables the best compared to all other potential variables available?

Other comments include implications, data and conclusions.

Why should this horizon be of interest to the reader? Moreover, could the pandemic of COVID invalidate his analysis?

In general terms, I believe that what is suggested could improve the readability of the manuscripts.

I suggest that the authors better specify, in a dedicated section, the current limits of the research and possible future developments, as I believe this topic is highly topical.

Thank you for the manuscripts.

Kind regards,

**Do you want your identity to be public for this peer review?** For information about this choice, including consent withdrawal, please see our Privacy Policy

Reviewer #1: No

Reviewer #2: **Yes: ** Thiago Coelho Soares

Reviewer #3: No

Reviewer #4: No

---

## [Author Response · Author response to Decision Letter 1]

2 Mar 2025

Response Reviewer 1

We are very grateful to the reviewers for their review and valuable comments on our papers. Your suggestions are crucial to improve the quality of the paper. We have made changes according to your comments and our response is as follows:

Commentator's Opinion:

a: Introduction The section lacks an outline of the article's marginal contributions and an overview of each section.

Author's Response:

We agree with this view and have added an outline of the marginal contribution of the article and a summary of each section to the introduction of the article. See page 5 for specific changes.

Commentator's Opinion:

b1: It is recommended that the background discussion should be extensive and the hypothesis development should be thoroughly examined through an extensive literature review. It is suggested to include references on government subsidies, such as: Han, Y., Han, L., Liu, C., & Wang, Q. (2024). How does government R&D subsidies affect enterprises’ viability? An investigation on inverted U-shaped relationship[J]. Finance Research Letters, 106235. https://doi.org/10.1016/j.frl.2024.106235

Author's Response:

The theoretical foundations section has been reorganised with relevant paragraphs based on your suggestions, and references to government subsidies (https://doi.org/10.1016/j.frl.2024.106235) have also been added, see the Theoretical Analysis and Research Hypotheses section on pages 6-11 for specific changes.

Commentator's Opinion:

b2: I did not see an in-depth analysis of the Theoretical Basis. It is recommended to add Institutional background.

Author's Response:

We agree with your suggestion for an in-depth analysis of the theoretical underpinnings, and we have reworked the Theoretical Analysis and Research Assumptions section in accordance with your suggestion, which can be found in the Theoretical Analysis and Research Assumptions section on pages 6-11. We believe the revised version is even better.

Commentator's Opinion:

c1: Please explain proprietary names such as ST, *ST, and PT to enable the reader to better understand their specific meaning, e.g. ST companies (Companies with special treatment due to two consecutive years of operating losses).

Author's Response:

We agree with this view and we have added an explanation of the proprietary name, as modified on page 12, line 238.

Commentator's Opinion:

c2: The authors selected China's A-share pharmaceutical listed enterprises from 2015 to 2022 as the research sample. It would be beneficial if the authors could explain further and justify the strong reasons for only using pharmaceutical listed enterprises from 2015 to 2022. Is there any possibility of bias? If so, how will the authors address this issue?

Author's Response:

We agree with this view. Firstly, the relationship between sustainable development and corporate innovation is one of the key concerns of pharmaceutical companies, so the main research target of this paper is pharmaceutical companies. Secondly, the rationale for using data from 2015 - 2022 has been added, as described in the Sample Selection and Data Sources section on pages 11-12. Finally, the article limitations have been added at the end of the article, with the opportunity to further explore other industries as research samples in the future, as shown on pages 44-45.

Commentator's Opinion:

d1: The choice of instrumental variables in the endogeneity discussion is not appropriate, and the results are not robust enough.

Author's Response:

We thank the reviewers for their suggestions regarding the endogeneity discussion. New references have been added, and the analysis of the KP-LM statistic shows that the probability of under-identification of the instrumental variables is significant and zero, which strongly rejects the original hypothesis of ‘unidentifiable’, confirms the correlation between the instrumental variables and the core explanatory variables, and shows that the instrumental variables selected in this paper are valid and reliable. See the discussion of endogeneity on page 23.

Commentator's Opinion:

d2: Although the article conducts mechanism examinations and heterogeneity analyses, these sections are too brief.

Author's Response:

We thank the reviewers for their suggestions to add content to the chapters on mechanism testing and heterogeneity analysis. Due to space constraints, we are unable to discuss this in detail in this paper. We plan to explore this issue further in future studies.

Response Reviewer 2

We are very grateful to the reviewers for their review and valuable comments on our papers. Your suggestions are crucial to improve the quality of the paper. We have made changes according to your comments and our response is as follows:

Commentator's Opinion:

1) Both ESG and enterprise innovation are latent variables, that is, it is not advisable to measure them by a single value. Thus, the appropriate statistical technique to understand the relationships between the variables and test the hypotheses would be through structural equation modeling and not regression. If a regression were to be used, it should be of the "Ordinal Regression" type, which would allow it to model the dependence of a polytomous ordinal response on a set of predictors, which can be factors or covariates. The Ordinal Regression design is based on the methodology of McCullagh (1980, 1998). This is because the ESG variable was collected in such a way as to be presented as an ordinary variable.

Author's Response:

We thank the reviewers for their suggestion regarding regression. However, due to limitations in ESG-related data collection, we are unable to implement this suggestion for the time being, while referring to the methodology of related studies (https://doi.org/10.1371/journal.pone.0301701;

https://doi.org/10.1371/journal. pone.0310447), we chose to keep the current methodology, which is related on pp. 14-15. We will attempt to address this issue in future research.

Commentator's Opinion:

2) Saying that ESG can be measured by "According to CSI ESG rating from low to high, the value is 1~9" is a very simplistic approach. In many studies, ESG is made up of more than a dozen variables. It is much more advisable for ESG to be a latent variable than an observable variable.

Author's Response:

We thank the reviewers for their suggestion regarding the measurement of ESG variables. Due to limitations in ESG-related data collection, we are unable to implement this suggestion at this time. In addition, we have added the addition of using this method to measure ESG in the Sample Selection and Data Sources section and in the Selection of Variables for ESG section, as modified on page 14, line 293. We will attempt to address this issue in future research.

Commentator's Opinion:

3) Converting ordinal variables into numerical variables is not appropriate. It is a statistically incorrect conversion and can generate distorted or unrealistic values. I understand that in PLOSONE alone there have been more than 16 articles in the last two years using this information incorrectly (https://doi.org/10.1371/journal.pone.0310447;
https://doi.org/10.1371/journal.pone.0299184;
https://doi.org/10.1371/journal.pone.0307535;
https://doi.org/10.1371/journal.pone.0295548;
https://doi.org/10.1371/journal.pone.0313686;
https://doi.org/10.1371/journal.pone.0301701, just to name a few). As an reviewer, I cannot help but point out that the conversion of ordinal variables into numeric variables and the statistical technique (regression for latent variables) used are incorrect and cannot be done in this way. The result generated ends up not having statistical validity. The correct thing to do would be to collect the variables in quantitative format and use structural equation modeling.

Author's Response:

We thank the reviewers for their suggestions regarding variable conversion. We have added citations of relevant literature using this method to measure ESG in the Variable Selection and Setting for ESG section as per your suggestion to use the article published in PLos one on variable measurement for ESG, see page 14, line 297 for the specific modification. We plan to make further attempts to address this issue in future studies.

Response Reviewer 3

We are very grateful to the reviewers for their review and valuable comments on our papers. Your suggestions are crucial to improve the quality of the paper. We have made changes according to your comments and our response is as follows:

Commentator's Opinion:

1. The abstract should be improved. Your findings can be expressed more clearly in the abstract, which should also briefly mention the policy implications.

Author's Response:

We agree with this view and have rephrased the findings in the summary section and added references to policy implications. See page 2 for specific changes.

Commentator's Opinion:

2. The study's aim and contribution should be expressed more strongly. The introduction needs to be more relevant to the discussion. The introduction's discussion and research contributions should be redesigned.

Author's Response:

We have reorganized the introductory and discussion-related paragraphs and believe that the revised version is even better. See pages 3-5 and 43-45 for specific changes.

Commentator's Opinion:

3. There needs to be more literature review and discussion of why this topic matters and how it fills a gap in the literature.

Author's Response:

We agree with this view and have added a further collation of the ESG related literature review in the introduction section. In addition, a discussion of the importance of green innovation for firms has been added. Finally, the marginal contribution of the article has been added to provide further discussion on filling gaps in the literature. See pages 3-5 for specific revisions.

Commentator's Opinion:

4. Control variables, such as PID (percentage of independent directors) and DUA (second job alignment), are typically used in corporate governance studies. They may need more relevance to the specific innovation outcomes being modeled.

Author's Response:

We thank the reviewers for raising the issue that individual control variables may need to be more relevant to the particular innovation being modelled. However, based on existing studies in the literature demonstrating the correlation of PID and DUA with firm innovation (https://doi.org/10.3390/ijerph19148558). Therefore, we retain the current choice of control variables. The details are as follows:

Two duty unification�DUA�: If the chairman and CEO are held by the same person, if the chairman and CEO are held by the same person, the corporate governance structure will be weakened, and the enterprise decision-making will lack democracy, which will have an adverse impact on green R&D.

Proportion of independent directors�PID�: When the proportion of independent directors is high, it indicates that the board of directors has strong independence, which can promote the improvement of enterprise performance and have a positive impact on green innovation.

Commentator's Opinion:

5. Some control variables, such as ROA, TQ, and LEV, might be highly correlated with each other, which could affect the stability of coefficient estimates in regression models.

Author's Response:

We thank the reviewers for raising the issue that some control variables can affect the stability of coefficient estimates in regression models. First, based on the existing literature (https://doi.org/10.3390/ijerph19148558;

https://doi.org/10.3389/fpsyg.2023.1096419;

Li Q, Xiao Z. Heterogeneous Environmental Regulation Tools and Green Innovation Incentives: Evidence from Green Patents of Listed Companies. Economic Research Journal. (2020)55: 192-208.) These control variables have a solid theoretical foundation; Second, we calculated the matrix of correlation coefficients between all variables (see S1 Table for details) and found that certain variables were indeed moderately correlated (with correlation coefficients between 0.4 and 0.6), but not to the extent of being highly correlated (>0.8); Finally, we further calculated the variance inflation factor (VIF), and the VIF values for all variables were less than 5, indicating that the multicollinearity problem was not serious. Therefore, we chose to keep these current control variables.

Commentator's Opinion:

6. There are typo mistakes and format issues. Such as line 212: “innovation inpu” and line 216: Error! Reference source not found.

Author's Response:

We agree with these suggestions of yours and have rechecked and corrected the typos and formatting issues.

Response Reviewer 4

Commentator's Opinion:

I suggest synthesizing the abstract, removing elements of detail, reviewing it following the IMRaD (Introduction-Methods-Results-Discussion) model, frequently used in empirical research in the natural and social sciences. In particular, the reference is to Methods M.

Author's Response:

We agree with you on this point and the abstract section has been reworked and a section on the methodological presentation of the article has been added. See page 2 for specific revisions.

Commentator's Opinion:

I would suggest presenting the objective now and better supporting the rest of the content, For example, some quotes could be added to support this: "The comprehensive advancement of China's manufacturing sector continues to demonstrate elevated energy consumption and significant pollution, accompanied by a low resource utilization rate resulting in the degradation of the ecological environment.".

Other suggestions for integration: formalize the research question and highlight the classic "Section' summary passage that closes introductory manuscripts.

Author's Response:

We agree with these suggestions of yours and have cited relevant literature to support the relevant content, see page 3, line 36 for specific changes.

We have followed your suggestions by providing paragraph summaries and research hypotheses in each subsection of the Theoretical Analysis and Research Hypotheses section. See pages 8-11 for specific changes.

Commentator's Opinion:

In "Research Design," the number of companies analyzed should be better specified. It is suggested to first, describe the variables used "RDit," "Patentit," and "ESGit," arguing concisely why they are the ones used and, useful for the research. Personal curiosity: why are these variables the best compared to all other potential; variables available?

Author's Response:

We agree with your suggestions.

Firstly, in the ‘Research design’, the article states the number of companies analysed, see line 242.

For firms‘ innovation investment measured by R&D investment, this paper refers to Shen Yi's study (https://doi:10.19616/j.cnki.bmj.2016.02.007.) The absolute value of R&D investment indicator, i.e., calculated as the natural logarithm of the amount of R&D investment can portray firms’ R&D behaviours in a more complete way. See line 280 for specific changes.

For the enterprise green innovation output measured by the number of green patents, firstly, the patent data can reflect a more specific output of the invention process, in addition, combined with China's information environment, the patent application indicator is more reflective of the enterprise's innovation research data. Therefore, this study refers to related research to measure green innovation output by the number of green patent applications of enterprises (https://doi.org/10.1016/j.iref.2024.103461). See line 285 for specific modifications.

For the measurement of ESG performance ability, we refer to related research methods (https://doi.org/10.1371/journal.pone.0301701;

https://doi.org/10.1371/journal.pone.0310447), and in addition, we combine China's national conditions and ESG-related data collection limitations to reflect the ESG performance of the sample companies. collection constraints, this paper adopts CSI ESG ratings to reflect the ESG performance of the sample companies. For details, please refer to line 292.

Commentator's Opinion:

Other comments include implications, data and conclusions.

Why should this horizon be of interest to the reader? Moreover, could the pandemic of COVID invalidate his analysis?

I suggest that the authors better specify, in a

---

## [Decision Letter · Decision Letter 1]

16 Apr 2025

Dear Dr. Chen,

We look forward to receiving your revised manuscript.

Kind regards,

Hafiz Muhammad Sohail, Phd

Academic Editor

PLOS ONE

Reviewers' comments:

Reviewer's Responses to Questions

**Comments to the Author**

Reviewer #1: (No Response)

Reviewer #5: All comments have been addressed

2. Is the manuscript technically sound, and do the data support the conclusions?

Reviewer #1: Partly

Reviewer #5: Yes

3. Has the statistical analysis been performed appropriately and rigorously?

Reviewer #1: N/A

Reviewer #5: Yes

4. Have the authors made all data underlying the findings in their manuscript fully available?

Reviewer #1: Yes

Reviewer #5: Yes

5. Is the manuscript presented in an intelligible fashion and written in standard English?

Reviewer #1: Yes

Reviewer #5: Yes

Reviewer #1: (No Response)

Reviewer #5: (No Response)

**Do you want your identity to be public for this peer review?** For information about this choice, including consent withdrawal, please see our Privacy Policy

Reviewer #1: No

Reviewer #5: **Yes: ** Isubalew Daba Ayana (PhD), Wollega University, Ethiopia

---

## [Author Response · Author response to Decision Letter 2]

1 May 2025

We agree with you on these suggestions. The relevant paragraphs of the introduction have been reorganized, and since the relationship between sustainability and corporate innovation is one of the key concerns of pharmaceutical companies, the main research target of this paper is pharmaceutical companies. Firstly, an explanation and discussion of the applicability of ESG performance to the pharmaceutical industry has been added to the introduction section, which is believed to be better in the revised version. See page 4 for specific modifications.

In addition, some of the article's limitations have been reorganized at the end of the article, with the main additions being the opportunity to further explore the differences in the impact of ESG performance on corporate innovation in pharmaceutical segments (e.g., biopharmaceuticals and traditional Chinese medicines, etc.) in the future, or to introduce a comparative study of other industries with the pharmaceutical industry. See page 45 for specific changes.

---

## [Decision Letter · Decision Letter 2]

19 Jun 2025

Dear Dr. Chen,

Thank you for submitting your manuscript to PLOS ONE. After careful consideration, we feel that it has merit but does not fully meet PLOS ONE’s publication criteria as it currently stands. Therefore, we invite you to submit a revised version of the manuscript that addresses the points raised during the review process.

We look forward to receiving your revised manuscript.

Kind regards,

Hafiz Muhammad Sohail, Phd

Academic Editor

PLOS ONE

Journal Requirements:

Reviewers' comments:

Reviewer's Responses to Questions

**Comments to the Author**

Reviewer #6: (No Response)

Reviewer #7: (No Response)

2. Is the manuscript technically sound, and do the data support the conclusions?

Reviewer #6: Yes

Reviewer #7: Yes

3. Has the statistical analysis been performed appropriately and rigorously?

Reviewer #6: Yes

Reviewer #7: I Don't Know

4. Have the authors made all data underlying the findings in their manuscript fully available?

Reviewer #6: Yes

Reviewer #7: (No Response)

5. Is the manuscript presented in an intelligible fashion and written in standard English?

Reviewer #6: No

Reviewer #7: Yes

Reviewer #6: (No Response)

Reviewer #7: The manuscript titled “The Impact of ESG Performance on Corporate Innovation: Empirical Evidence from Chinese Pharmaceutical Listed Companies” examines the relationship between Environmental, Social, and Governance (ESG) performance and corporate innovation within China's pharmaceutical industry. Utilizing panel data from 2015 to 2022, the study employs fixed-effects and mediation models to analyze both innovation input and output, with R&D personnel and government subsidies serving as mediating variables. The inclusion of system GMM and robustness tests, such as quantile and Poisson regressions, further strengthens the reliability of the findings.

One of the key strengths of the study is its focus on an emerging and highly relevant research area—the intersection of ESG sustainability and corporate innovation, particularly within the impactful pharmaceutical sector. The research is well-structured, policy-relevant, and timely, offering actionable insights. For instance, the findings suggest the implementation of ESG-based incentive policies and more strategic allocation of government subsidies to foster innovation. These recommendations could have significant implications for policymakers and industry stakeholders.

A minor language improvement was noted: the phrase "advantageously correlated" could be replaced with "positively associated" for greater clarity and precision. Overall, the manuscript makes a valuable contribution to the literature by providing empirical evidence on how ESG performance drives innovation, supported by robust methodological approaches. The study’s insights are particularly pertinent for fostering sustainable growth in the pharmaceutical industry.

**Do you want your identity to be public for this peer review?** For information about this choice, including consent withdrawal, please see our Privacy Policy

Reviewer #6: **Yes: ** Nadeem Akhtar Khan (School of Economics & Finance Xi'an Jiao tong University Xi'an China)

Reviewer #7: No

---

## [Author Response · Author response to Decision Letter 3]

2 Sep 2025

We are very grateful to the reviewers for their review and valuable comments on our papers. Your suggestions are crucial to improve the quality of the paper. We have made changes according to your comments and our response is as follows:

Commentator's Opinion:

Title and Abstract: The title accurately reflects the content. However, consider rephrasing for smoother grammar: "Empirical Evidence from Chinese Pharmaceutical Listed Companies".

Author's Response:

We agree with you on these suggestions. We have modified the title in accordance with the suggested changes.See page 1 for specific modifications.

Commentator's Opinion:

Introduction: The introduction is thorough and offers good context on ESG policy developments in China. However, the theoretical rationale is fragmented.

Author's Response:

We agree with your suggestion.We considered that placing the theoretical basis in the introduction section would make that section too lengthy, so we moved the content related to the theoretical basis of ESG and corporate innovation to the second section, ‘Theoretical basis and research hypothesis.’。See page 6-7 for specific modifications.

Commentator's Opinion:

Methodology:Clearly explain the rationale behind using CSI ESG ratings and their scoring system. A table detailing ESG score distribution across years/firms could help.

Author's Response:

We agree with you on these suggestions.We added descriptive statistics for the ESG scores of the research samples from 2015 to 2022 and conducted a descriptive analysis of the statistical results.See page 18-20 for specific modifications.

In addition, descriptive statistics tables and DO files for corporate ESG scores from 2015 to 2022 have been re-uploaded to the data management platform to improve the reproducibility of the results. The specific path is as follows:

https://data.mendeley.com/preview/4r554m7kjm?a=e55a37e4-5626-491d-bd2b-a7558a382ce0

Commentator's Opinion:

Data and Variables:

The innovation input (log R&D expenditure) and output (green patents) are standard but consider briefly addressing limitations of patent counts (e.g., time lags, quality variation).

The variable descriptions are useful but could be more reader-friendly if organized in a summary table with examples.

Author's Response:

We agree with your suggestion.

First, we conducted an in-depth consideration and relevant tests on the issue of patent count restrictions in the robustness test section.On the one hand, considering the time lag effect of corporate ESG performance on corporate innovation, this study uses the lagged value of the explanatory variable for regression analysis. On the other hand, considering that the number of patent applications is a count variable, using only a fixed-effects model for regression analysis may not be robust enough. Therefore, this study re-conducts empirical tests using a Poisson regression model to verify the correlation between the main research variables.See page 24-26 for specific modifications.

Second, we summarized all variables in tabular form, as shown in Table 1.See page 16-18 for specific modifications.

Commentator's Opinion:

Language and Style:

The manuscript has many grammatical and stylistic issues (e.g., awkward phrasing, inconsistent tense, punctuation). A professional English editing service is recommended.

Tables are dense and occasionally unclear. Consider splitting long tables or highlighting key coefficients.

Author's Response:

We agree with your suggestion.

First, we re-checked and revised the wording and tense of the entire text.

Second, in response to the issue of dense tables, we deleted the data display of control variables and uploaded all complete tables as supporting information.

---

## [Editor Report · Decision Letter 3]

8 Sep 2025

The Impact of ESG Performance on Corporate Innovation -Empirical Evidence from Chinese Pharmaceutical Listed Companies

PONE-D-24-54146R3

Dear Dr. Chen,

We’re pleased to inform you that your manuscript has been judged scientifically suitable for publication and will be formally accepted for publication once it meets all outstanding technical requirements.

Kind regards,

Hafiz Muhammad Sohail, Phd

Academic Editor

PLOS ONE
---

## [Editor Report · Acceptance letter]

PONE-D-24-54146R3

PLOS ONE

Dear Dr. Chen,

I'm pleased to inform you that your manuscript has been deemed suitable for publication in PLOS ONE. Congratulations! Your manuscript is now being handed over to our production team.

Kind regards,

on behalf of

Dr. Hafiz Muhammad Sohail

Academic Editor

PLOS ONE